# Chemical evolution of primordial salts and organic sulfur molecules in the asteroid 162173 Ryugu

Toshihiro Yoshimura [1,40] ✉, Yoshinori Takano [1,40], Hiroshi Naraoka[2], Toshiki Koga [1], Daisuke Araoka [3], Nanako O. Ogawa [1], Philippe Schmitt-Kopplin [4,5], Norbert Hertkorn [4], Yasuhiro Oba [6], Jason P. Dworkin [7], José C. Aponte [7], Takaaki Yoshikawa[8], Satoru Tanaka[9], Naohiko Ohkouchi[1], Minako Hashiguchi [10], Hannah McLain[7], Eric T. Parker[7], Saburo Sakai[1], Mihoko Yamaguchi[11], Takahiro Suzuki[11], Tetsuya Yokoyama [12], Hisayoshi Yurimoto [13], Tomoki Nakamura[14], Takaaki Noguchi [15], Ryuji Okazaki[2], Hikaru Yabuta [16], Kanako Sakamoto [17], Toru Yada [17], Masahiro Nishimura[17], Aiko Nakato [17], Akiko Miyazaki [17], Kasumi Yogata [17], Masanao Abe[17], Tatsuaki Okada [17], Tomohiro Usui [17], Makoto Yoshikawa[17], Takanao Saiki[17], Satoshi Tanaka[17], Fuyuto Terui[18], Satoru Nakazawa [17], Sei-ichiro Watanabe [10], Yuichi Tsuda[17], Shogo Tachibana [17,19] & Hayabusa2-initial-analysis SOM team*

Samples from the carbonaceous asteroid (162173) Ryugu provide information on the chemical evolution of organic molecules in the early solar system. Here we show the element partitioning of the major component ions by sequential extractions of salts, carbonates, and phyllosilicate-bearing fractions to reveal primordial brine composition of the primitive asteroid. Sodium is the dominant electrolyte of the salt fraction extract. Anions and $NH_4^+$ are more abundant in the salt fraction than in the carbonate and phyllosilicate fractions, with molar concentrations in the order $SO_4^{2-} > Cl^- > S_2O_3^{2-} > NO_3^- > NH_4^+$. The salt fraction extracts contain anionic soluble sulfur-bearing species such as $S_n$-polythionic acids ($n < 6$), $C_n$-alkylsulfonates, alkylthiosulfonates, hydroxyalkylsulfonates, and hydroxyalkylthiosulfonates ($n < 7$). The sulfur-bearing soluble compounds may have driven the molecular evolution of prebiotic organic material transforming simple organic molecules into hydrophilic, amphiphilic, and refractory S allotropes.

The Hayabusa2 spacecraft provided the opportunity to investigate the carbonaceous parent body and the astrochemical record of the carbonaceous asteroid (162173) Ryugu[1–3]. The soluble organic matter (SOM) and soluble ionic compositions of Ryugu have been attributed to prebiotic molecular evolution with unique aqueous alteration effects on the parent asteroid[4]. Furthermore, the characterization of organic-rich carbonaceous chondrites and their classification into groups, such as the Mighei- (CM), Renazzo- (CR), and Ivuna-type (CI) groups, has improved understanding of the origin of water-bearing aqueous alteration minerals and their association with organic molecules. This has provided important insights into the primary chemical profiles and duration of aqueous alteration in the early

A full list of affiliations appears at the end of the paper. *A list of authors and their affiliations appears at the end of the paper.
✉ e-mail: yoshimurat@jamstec.go.jp

solar system e.g.[5–8]. The chemical composition of the samples recovered from Ryugu has been found to be most similar to that of the CI group[9].

Soluble ions act as bulk electrolytes that stabilize surface charge, and they potentially have a specific structural role in organic and inorganic molecules. In addition, because astrochemically relevant volatile and nonvolatile organic molecules may be present both as salts and in bound forms (i.e., physically trapped or chemically bonded to the matrix), they are less likely to be lost by evaporation from the parent body e.g.[10]. Ultra-high-resolution mass spectrometry has revealed that various soluble CHO, CHNO, CHOS, and CHNOS species, as well as organometallic CHO−Mg species, are present in the SOM of meteorites[11,12]. The overall compositional diversity of organic molecules in Murchison meteorite extracts surpasses the compositional diversity of terrestrial biochemical organic matter[12]. The order of molecular compositional diversity of Murchison solvent extracts, CHNOS > CHNO > CHOS > CHO, indicates a significant contribution of sulfur. Furthermore, differences in the distributions of mass peaks between CHOS and CHNOS molecules with average H/C ratios imply divergent formation pathways and the loss of precursor signatures of source CHNO and CHO materials[12]; however, the formation mechanisms of CHOS and CHNOS molecules are not yet fully understood. A recently observed high thermal stability of sulfur-magnesium-carboxylates (CHOSMg) may contribute to the survival of organic molecules under harsh extraterrestrial conditions[13]. Soluble ions may contribute to the stabilization of organic matter by forming complexes or by existing as salts; however, basic issues such as the composition and charge balance of ions that leach out with SOM have not yet been investigated. Ryugu samples provide data on the astrochemical history of pristine organic matter and its chemical environment.

Initial analysis of organic molecules in samples retrieved from the surface of the C-type asteroid Ryugu has revealed a high molecular diversity of CHNOS species[4,14–17]. A variety of organic compounds, such as racemic mixtures of proteinogenic and nonproteinogenic amino acids, aliphatic amines, carboxylic acids, polycyclic aromatic hydrocarbons, and nitrogen-containing heterocyclic compounds, has been detected, suggesting that long-term chemical processes caused by aqueous alterations may have contributed to the prebiotic molecular evolution of Ryugu[4]. The addition of sulfur functionalities onto CHNO and CHO precursor molecules might have occurred during interactions mediated by water or during solid-state reactions even under the mild temperature conditions of Ryugu (i.e., fluid alteration at $37 \pm 10\,°C$, never heated above ~100 °C after aqueous alteration[9]). However, soluble sulfur species available to effect S-functionalization are presently unaccounted for in the primordial formation and molecular evolutionary histories of organosulfur compounds.

We report here the distribution patterns of major cations, anions, and sulfur compounds in the salt-, carbonate-, and phyllosilicate-bearing fractions of two surface samples from Ryugu (A0106 from the first touchdown site, and C0107 from the second touchdown site). Soluble sulfur compounds were identified, which could have been intermediate reactive species in the primordial organic and inorganic molecular evolution on Ryugu. In addition, sulfur has been proposed to exist in the solar system as allotropes such as $S_8$ (cyclooctasulfur)[14]. These sulfur species are almost impossible to observe via astronomical spectroscopy, but cosmic-ray-driven radiation has recently been proposed as a mechanism for the formation of sulfur allotropes[18]. We also discuss the process by which highly reactive sulfur is transformed into a more stable chemical form. Furthermore, we quantify the major soluble components of representative reference carbonaceous meteorites (CI1 Orgueil, C2$_{ung}$ Tarda, and CM2 Aguas Zarcas and Jbilet Winselwan) to compare the distribution of the major soluble components over a range of aqueous alteration.

## Results

### Total sulfur content and isotopic profiles

The total sulfur content (S, wt%) and sulfur isotopic composition ($\delta^{34}S$, ‰ vs. Vienna Canyon Diablo Troilite [VCDT]) of the studied Ryugu samples (A0106 and C0107) and of representative carbonaceous meteorites are shown in Fig. 1A. Sulfur abundance differed between A0106 ($3.3 \pm 0.7$ wt%)[4] and C0107 ($5.5 \pm 0.7$ wt%, $n = 5$; sample weight = $20.6 \pm 5.4\,\mu g$), implying a heterogeneous distribution of sulfides, either horizontally, between the touchdown 1 and 2 sampling locations, or vertically, between the surface and subsurface samples[1,19,20] (Supplementary Table 1). The bulk sulfur isotopic compositions of the Ryugu samples, however, indicated a homogenous distribution: $\delta^{34}S = -3.0‰ \pm 2.3‰$ for A0106 and $\delta^{34}S = -1.10‰ \pm 1.62‰$ ($n = 5$) for C0107.

### Major cations and anions in the Ryugu extracts

We quantified the anions and cations present in the hot ultrapure $H_2O$-, HCOOH-, and HCl-soluble phases of the Ryugu samples (Supplementary Figs. 1–3). In these fractions, salts, carbonates, and phyllosilicates are the primary host solids. Hereafter, the fraction comprising anionic and cationic components extracted by hot ultrapure water is defined as the salt fraction (see the caption of Fig. 1B for the original fraction numbers associated with the different solids[4]). A major cation in the salt fraction of Ryugu is $Na^+$, which acts as a primary bulk electrolyte (Figs. 1B and 2). In addition, thiosulfate is found only in the $H_2O$-soluble form. A small amount of $NH_4^+$ was also detected in this fraction of Ryugu (0.18 μmol/g for the A0106 salt fraction and <0.02 μmol/g for C0107): Orgueil contains ~200 times as much $NH_4^+$ (33.54 μmol/g, see Supplementary Information). Anions were more concentrated in the salt fraction than in the carbonate and phyllosilicate fractions, with molar concentrations in the following order: $SO_4^{2-} > Cl^- > S_2O_3^{2-} > NO_3^-$ (Supplementary Table 2). Unsurprisingly, the organic fraction of all the samples that was extracted by dichloromethane/methanol (DCM/MeOH) contained fewer ionic solutes because of its low polarity compared to ultrapure $H_2O$, HCOOH, and HCl.

### Soluble sulfur-containing compounds

Thiosulfate accounts for 43% of the total dissolved S in the salt fractions from both A0106 and C0107 (Supplementary Table 2). Polythionates yielded the most intense signals in the ultra-high-resolution mass spectra, and were also detected in the methanol extract[4] (extract #4, Supplementary Fig. 4). Sulfuric acid accounts for approximately 6.3% of the total sulfur in A0106 and C0107; thiosulfate makes up approximately 2.4% of the total sulfur in A0106 and C0107 (Supplementary Table 3).

The ion species listed in Supplementary Table 2 were measured by ion chromatography with an electrical conductivity detector. For compounds requiring precise mass, we used ion chromatography high-mass-resolution spectrometry (IC-Orbitrap-MS) to detect a series of anion species in the mass range $m/z$ 40–750. Sulfur-bearing species were the main components of the dissolved anions in the Ryugu salt fraction (Fig. 3, Supplementary Fig. 5). In addition to sulfuric and thiosulfuric acid, quantified by IC in conductivity detection mode, homologous molecules of $S_n$-polythionic acids ($n < 6$) and $C_n$-hydroxyalkylsulfonates ($n < 7$, Appendix) were detected in a Ryugu salt fraction recovered by using an extraction order different from that used for the other salt fractions (extract #5; Supplementary Figs. 1B and 5). The $pH$ of the salt fraction (#7-1 in Supplementary Fig. 1B) measured at 24.6 °C was weakly acidic, with values of $3.946 \pm 0.004$ for A0106 and $4.186 \pm 0.006$ for C0107 during the multistep scanning (Supplementary Fig. 6).

## Discussion

Large variations of element abundances were observed between the salt and carbonate fractions of Ryugu (Fig. 1). In the salt fraction, $Na^+$ accounted for ~90% of the cations (Fig. 2, Supplementary Table 2). Water-chemistry modeling of Ryugu[21] indicated high Na concentrations

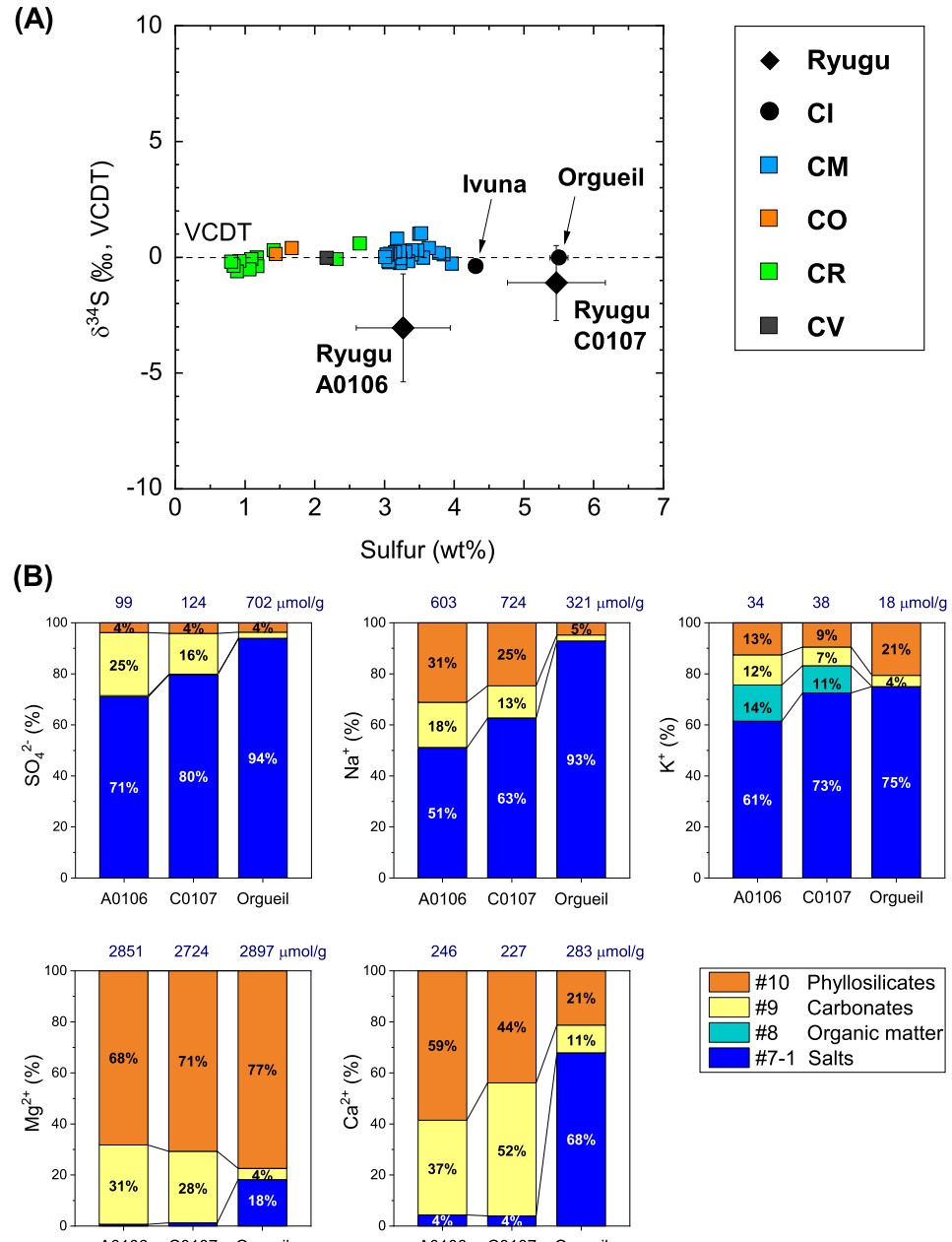

**Fig. 1 | Element compositions of the Ryugu samples. A** Total sulfur content (wt%) and isotopic profiles of the Ryugu samples (A0106, C0107) and of representative carbonaceous groups (CI, CM, CO, CR, and CV). Sulfur (S, wt%) and δ34S (‰ vs. VCDT) values are from the literature[4,17,47] and references therein. Error bars are one standard deviation (1 SD) values of multiple particles. **B** Relative amounts of sulfate, sodium, potassium, magnesium, and calcium in sequential solvent extracts of the samples collected at the first touchdown site (A0106) and the second touchdown site (C0107) on the asteroid Ryugu (Supplementary Fig. 1), and in a sample from Orgueil (values less than 3% were omitted). C0107 may contain subsurface samples

from ejecta associated with the artificially made impact crater. We used fine-grained samples and carried out the sequential solvent extraction in a clean room[4] (Supplementary Fig. 1). We measured evaporitic salts (via #7-1 hot water extraction, see IDs in Supplementary Fig. 1 and Naraoka et al.[4]); ions bound to soluble organic matter (via #8 dichloromethane and methanol, DCM+MeOH); exchangeable ions and highly soluble minerals such as carbonates (via #9 formic acid, HCOOH); and clays and residual soluble minerals (via #10 hydrochloric acid, HCl). Navy numbers are the sum of extractable solute contents for each solute. Data are provided as a Source Data file.

at low water-to-rock ratios in both fluid and saponite, consistent with the Na-rich composition of the least-altered lithology of Ryugu. The model demonstrates an evolution from Mg–Na–Cl solutions in the early stages of aqueous alteration toward Na–Cl alkaline brines. A striking difference between the modeled fluid and the Ryugu extracts was observed in the profiles of soluble oxygenated sulfur species (Fig. 3). The observed high molecular diversity of S-bearing species provides information about the formation processes of these compounds in Ryugu, as discussed below. The major extractable solutes from the salt and carbonate fractions

accounted for 8.0 wt% of solid materials in both A0106 and C0107 (Supplementary Table 4); these solutes are considered to have been the major solutes in the Ryugu brine at the onset of salt desiccation. During aqueous alteration, water is consumed by competing hydration and oxidation reactions[22]. A decrease in the water content would lead to the precipitation of inorganic evaporite minerals. However, inorganic salts (especially Na-, Cl-, and S-bearing salts) are present in only trace amounts or are absent in the Ryugu particles[21], and in polished sections, carbonates are seen to consist of acid-soluble dolomite [Ca, Mg(CO3)2],

breunnerite [(Mg, Fe)CO$_3$], and calcite (CaCO$_3$), with none of the identified highly soluble Na-containing minerals that may be present in the salt fraction[9,21]. During aqueous alteration, the major solutes interact with the organic matter, followed by insertion reactions and complexation to SOM. The cation excess (Supplementary Table 4) of our sequential extracts can be considered to represent a balance between SOM (including R–SO$_3^-$ and R–OSO$_3^-$; Figs. 3, 4), dissolved inorganic carbon species, and dissolved silica. The absence of inorganic sulfate salts has been mineralogically confirmed[9,21]. Anion adsorption on clay minerals is unlikely to result in large amounts of dissolved sulfur, because saponite has a predominantly negative surface charge[23]. Therefore, it is likely that the anions are stored as functional groups of soluble organic matter or their salts. Organosulfur-bearing anions are regarded as credible counterions to Na$^+$ in the early solar system; aliphatic amines have been proposed to be present as affinity salts in the grains[24]. We compared the concentrations of soluble ions available for organic reactions with the solvent solubility parameter for hydrophobicity and hydrophilicity of each organic solvent (Fig. 5). The increase in the solubility parameter with increasing cation concentrations indicates that the amount and distribution of ions available for organochelates is essentially a function of polarity.

Sulfate salts were not detected in Ryugu samples during non-destructive microscopic observation[9,21]. Those previous studies concluded that the sulfate veins in CI meteorites are products of weathering on Earth. In addition, the chemical composition of the Ryugu particulate matrix was different from that of CI Orgueil, and sulfate and ferrihydrite were formed from sulfides in CI[25]. The abundant water-soluble Mg sulfate veins (epsomite, MgSO$_4$·7H$_2$O) in CI Orgueil[26] may be a primary desiccation product in the parent asteroid or represent remobilization and reprecipitation of soluble sulfates resulting from interaction with the Earth's atmosphere[27,28]. In the Mg–Ca–Na+K diagram, the salt fraction of Orgueil plots near the carbonate fraction (Fig. 2); in addition, these fractions have exceptionally similar $\delta^{26}$Mg values (Supplementary Fig. 7). The significant enrichment of sulfate in the Orgueil salt fraction (Fig. 1B) cannot be explained by redistribution of sulfate from the carbonate and phyllosilicate fractions (Supplementary Table 2); therefore, the observed sulfate enrichment is considered to derive from the oxidation of FeNi sulfides. Thus, terrestrial alteration processes such as moisture absorption and the consequent precipitation have potentially modified CI-type meteorites from their primary composition cf.[9]. Magnesium sulfates are characterized by very high solubility and rapid dissolution in water due to their hygroscopic nature even under atmospheric conditions; thus, in evaporite formations on Earth, they are a precipitate indicating very high salinity[29]. Although the terrestrial origin of Mg sulfates has been scrutinized from both historical and scientific perspectives[28], the present $\delta^{26}$Mg result for the Orgueil salt fraction indicates that Mg was redistributed mainly from dolomite. Hence, when Mg-bearing carbonates and silicate fractions dissolve during terrestrial alteration processes, the Mg should be partitioned to the salt fraction as a concentrate product. Analysis results of both elemental concentrations and isotopic ratios support the redistribution of Mg in Orgueil, confirming the chemically pristine nature of the Ryugu samples[9].

Polythionates are often produced by thiosulfate oxidation, but there are as yet few constraints on sulfur chemistry and the formation of organosulfur compounds in carbonaceous chondrite parent bodies[30]. Polythionates are stable under acidic conditions ($p$H ca. 4–5)[31], consistent with the measured $p$H of the salt fractions of both A0106 and C0107 (Supplementary Fig. 6). A generalized reaction for aqueous polythionate formation has been suggested as follows[32] [Eq. 1]:

$$(2n-5)H_2S + (n+5)SO_2 \rightarrow 3\,S_nO_6^{2-} + 6\,H^+ + (2n-8)H_2O \quad (1)$$

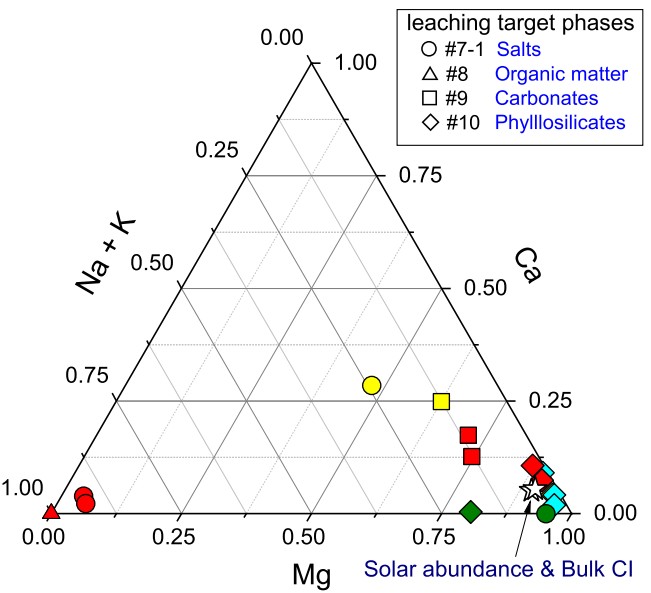

**Fig. 2 | Ternary diagram illustrating the molar proportions of Mg, Ca, and Na + K in the sequential extracts.** Samples include Ryugu A0106 and C0107 (red), Orgueil (yellow), Tarda, Aguas Zarcas, Jbilet Winselwan (blue), and serpentine (olive), with the bulk compositions of CI chondrite and solar abundance (stars) also plotted for ref. 48. The types of solvents and the main target phases of the leaching experiments are documented in Supplementary Fig. 1B. Data are provided as a Source Data file.

Chemical equilibrium modeling of aqueous alteration of the Ryugu parent body, with mixing of rocks, water containing CO$_2$ and HCl, and organic matter, yields low Water/Rock ratios (W/R, ranging from 0.06–0.1 for least-altered to 0.2–0.9 for extensively altered lithologies) and high Na concentrations in both fluids and the secondary mineral saponite[21]. Previous comprehensive thermodynamical modeling has demonstrated that, under low W/R conditions, neutralization of the initial HCl-containing acidic solution results in a Na-rich alkaline fluid in which Na-containing secondary minerals such as saponite can stably exist[22]. The Ryugu results suggest the existence of Mg–Na–Cl-rich solutions in the early stages of aqueous alteration, which evolved into more reductive, Na–Cl alkaline brines that coexisted with H$_2$-rich gas phases[21]. The solutes with high solubility extracted by our hot H$_2$O method also yield a Na-rich composition, consistent with the chemical modeling. The most likely possibility may be that polythionate formation occurred by solid-phase reactions after the escape of reducing substances such as H$_2$ and CH$_4$. Note that the weak $p$H acidity of the eluate of the salt fraction is based on the ionic balance, which is influenced by both primary aqueous alteration and subsequent molecular evolution. For example, the Ryugu SOM contains monocarboxylic acid[4], which is weakly acidic, and we consider that the $p$H of the salt fraction also reflects this SOM characteristic. The subsequent elongation of the polythionate chain and the formation of thiosulfate are governed by the reaction shown in Eq. 2[33], and then a hydrolysis reaction (Eq. 3 shows that for tetrathionate, but the reactions for other S$_n$O$_6^{2-}$ are analogous) produces S$_8$ as the most stable end product[34] (S$_8$ detection by Aponte et al. 2023[14]) [Eq. 3]:

$$S_{n+1}O_6^{2-} + SO_3^{2-} \leftrightarrow S_nO_6^{2-} + S_2O_3^{2-} \quad (2)$$

$$8\,S_4O_6^{2-} + 8\,H_2O \rightarrow 8\,S_2O_3^{2-} + S_8 + 8\,SO_4^{2-} + 16\,H^+ \quad (3)$$

These reactions are summarized in Fig. 4, along with the oxidation state of sulfur. In the particle-phase reaction, high-molecular-weight

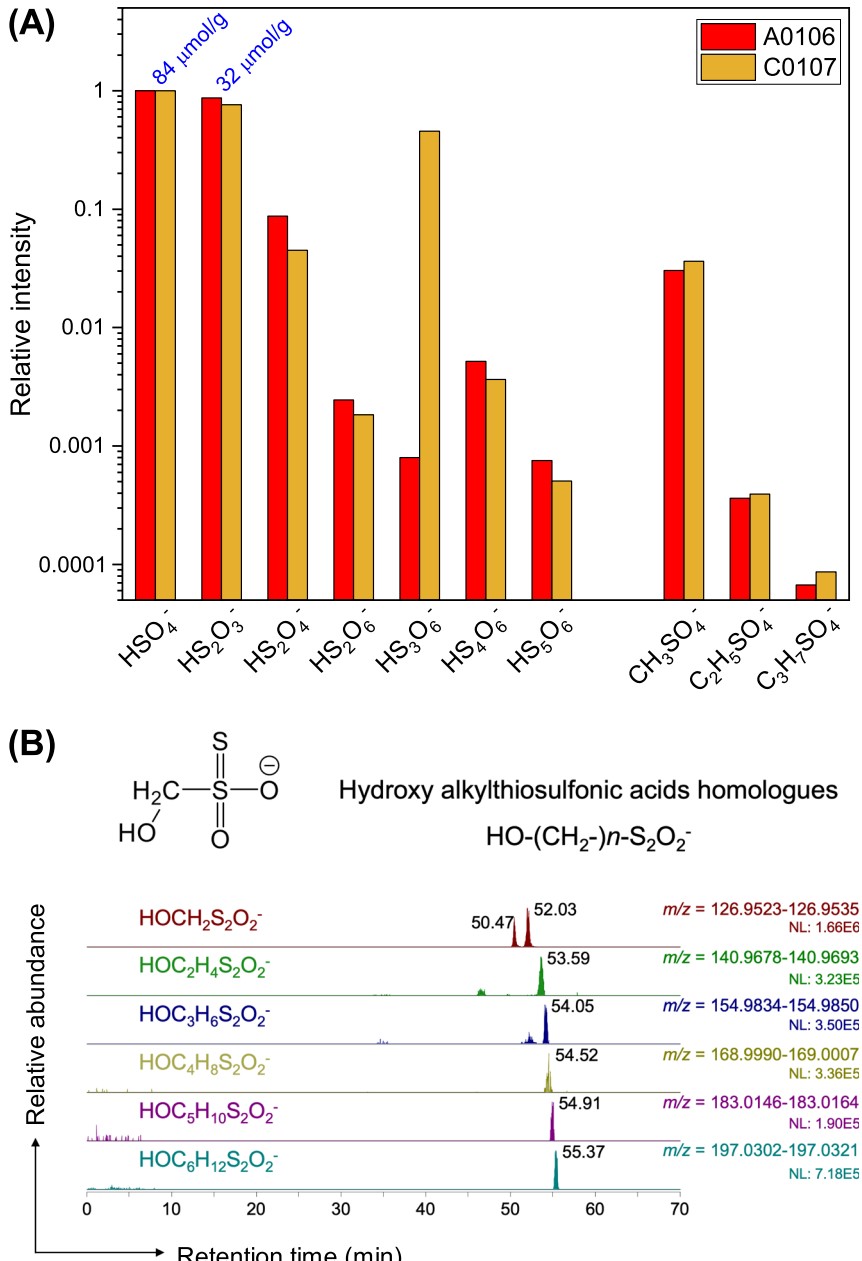

**Fig. 3 | Anionic soluble sulfur-bearing species. A** Relative intensity of major anion species detected by ion chromatography / Orbitrap mass spectrometry (IC/Orbitrap MS) of the water extracts of A0106 and C0107 (#5, Supplementary Fig. 1B). The intensities are normalized to sulfuric acids and shown on a logarithmic scale. Average sulfate and thiosulfate concentrations of A0106 and C0107 quantified by conductivity detection of ion chromatography analysis are shown above the bars (blue). **B** Representative nano-flow LC/Orbitrap MS chromatograms of water-extractable (#5) organic sulfur homologs. Here we show hydroxy alkylthiosulfonic acid HO−$(CH_2$−)$n$−$S_2O_2^-$ obtained from Ryugu sample A0106. Other organosulfur compounds are shown in Supplementary Figs. 4 and 8. Data are provided as a Source Data file.

organosulfur species are formed by oligomerization under various surface acidities and oxidative conditions[35]. Furthermore, the variability of the composition of the organosulfur compounds increases when the particle-phase acidity is derived from sulfonate groups[35]. Thus, the presence of soluble oxygenated sulfur species and the resulting weakly acidic conditions provide an opportunity for extending the high molecular diversity of extraterrestrial CHNOS and CHOS molecules. It is also possible that intermediate redox products (e.g., $H_2O$ and $CO_2$) produced during the reactions contributed to the water−mineral reaction of inorganic minerals, consistent with the detection of $CO_2$-bearing aqueous fluid[21].

Another important property of sulfonate groups is their hydrophilic nature, which results in the molecular variation trends of hydrophilic molecules observed in the Murchison meteorite: CHNOS > CHNO > CHOS > CHO[12]. Our analysis indicates that sulfate esters (R−O−$SO_3^-$: $CH_3SO_4^-$, $C_2H_5SO_4^-$, and $C_3H_7SO_4^-$) are the most abundant organosulfur compounds in the salt fraction. These compounds can be formed by sulfate esterification [Eq. 4]:

$$R − OH + HSO_4^{\;-} \leftrightarrow R − OSO_3^{\;-} + H_2O \qquad (4)$$

## Number of sulfur atom

| Average formal charge on sulfur | 1 | 2 | 3 | 4 | 5 | 6 | 7 | 8 |
|---|---|---|---|---|---|---|---|---|
| -II | $S^{2-}$ | | | | | | | |
| -I | | $S_2^{2-}$ | $S_3^{2-}$ | $S_4^{2-}$ | $S_5^{2-}$ | $S_6^{2-}$ | $S_7^{2-}$ | |
| 0 | | $S_n$ | | | | | | $S_8$ |
| +I | | | $S_3O_3^{2-}$ | $S_4O_3^{2-}$ | $S_5O_3^{2-}$ | $S_6O_3^{2-}$ | $S_7O_3^{2-}$ | |
| +II | | $S_2O_3^{2-}$ | | $(S_nO_6^{2-})$ $S_4O_6^{2-}$ | $S_5O_6^{2-}$ | $S_6O_6^{2-}$ | $S_7O_6^{2-}$ | |
| +III | | $S_2O_4^{2-}$ | $S_3O_6^{2-}$ | | | | | |
| +IV | $SO_3^{2-}$ | $S_2O_5^{2-}$ | | | | | | |
| +V | | $S_2O_6^{2-}$ | | | | | | |
| +VI | $SO_4^{2-}$ | $S_2O_7^{2-}$ | | | | | | |
| +VII | | $S_2O_8^{2-}$ | | | | | | |

Organo-S compounds: $R\text{-}SO_3^-$, $R\text{-}OSO_3^-$ (Eq. 5)

**Fig. 4 | Reactions along with the oxidation state of sulfur.** Sulfur species and reaction pathways described by Eqs. 1–4 (after Williamson and Rimstidt, 1992)[49]. Two dominant species of organosulfur compounds ($R\text{-}SO_3^-$ and $R\text{-}OSO_3^-$) were reported previously[36] and are documented in this study. The red line shows the reaction path from inorganic ions to these sulfur-containing organics, such as esterification (Eq. 4). The purple line shows the reaction path of sulfur allotropes stabilizing to S8. Sulfur species detected by our ion chromatography and mass spectrometry analyses are shown in the orange squares. Note that organosulfur compounds with various alkyl side chains have been detected[4,14] (Fig. 3, Supplementary Fig. 8). For Eq. 1, the presence of $SO_2$ has been suggested by spectroscopic observations[50], and both $H_2S$ and $SO_2$ are generally involved in astrochemical models[18]. See also Supplementary Tables 1, 3 and their references for sulfur abundance in the Ryugu sample.

In addition, hydroxyalkylsulfonic acid can be formed by the the formose reaction of aldehydes quenched by bisulfite[36] [Eq. 5]:

$$R-CHO + HSO_3^- \leftrightarrow R-CH(OH)SO_3^- \tag{5}$$

Organosulfate compounds can be amphiphilic because they are composed of a hydrophilic sulfate group and a hydrophobic hydrocarbon group; thus, they could accumulate on particle surfaces and facilitate inorganic–organic interactions. It is conceivable that such amphiphiles could leave remnant structures, such as the organic nanoglobules observed in carbonaceous meteorites[37]. Isotopic studies such as those carried out on meteorites[38] could clarify the role of this potential mechanism of nanoglobule formation in Ryugu. During further reactions, a variety of soluble organosulfur species are formed (Fig. 4). Alkyl sulfonates produced through the reaction of HCHO and $HSO_3^-$ have been identified in methanol extracts of the Murchison, Tagish Lake, and Allende meteorites[37]. In the Ryugu methanol extracts, higher-carbon-number species of (hydroxy-)alkylsulfonic/alkylthiosulfonic acid homologs are present (#4, Fig. 3B, Supplementary Fig. 8), as well as sulfide species such as dimethyl disulfide, dimethyl trisulfide, and dimethyl tetrasulfide[14]. During self-assembly and transformation of molecules into prebiotic building-blocks, amphiphilic compounds play a key role in encapsulating or integrating insoluble macromolecular matter[39] such as mono- and polyaromatic units bonded with O and S and the small aliphatic chains commonly found in carbonaceous chondrites.

Finally, the polythionate undergoes decomposition to form a lower-sulfur-number polythionate with elemental sulfur ($S_n$) [Eq. 3].

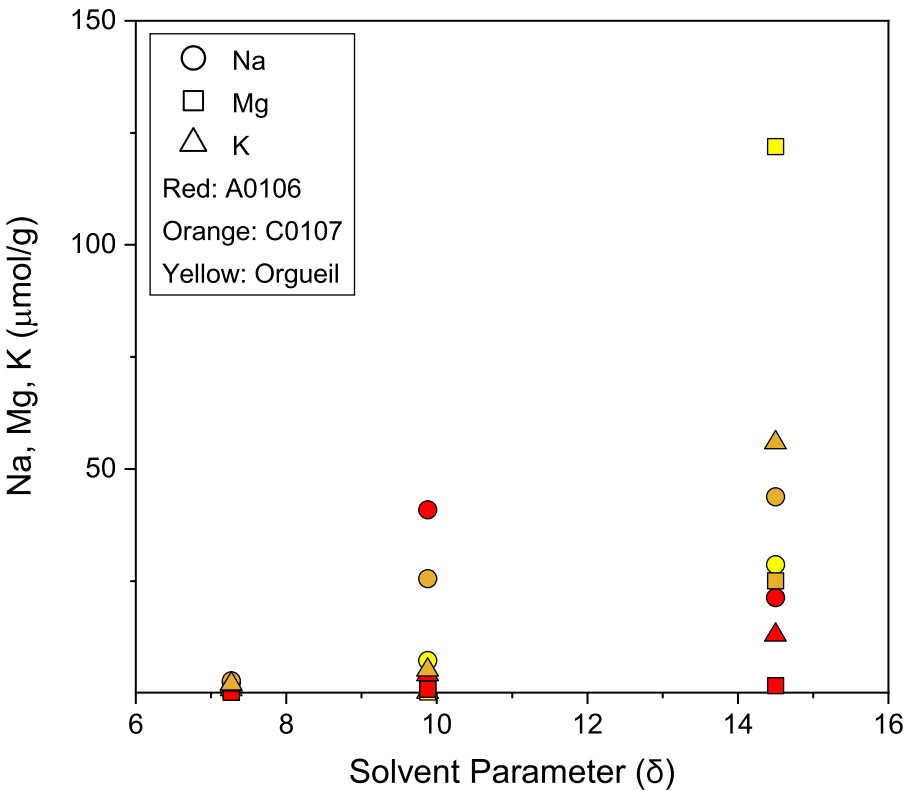

**Fig. 5 | Sodium (Na), magnesium (Mg), and potassium (K) concentrations in sequential organic solvent extracts vs. solvent solubility parameters (δ)[51].** The solutes were extracted sequentially from lower to higher δ values[4]; hexane (δ = 7.3), dichloromethane (δ = 9.9), and methanol (δ = 14.5). For reference, the δ value of water is 23.5. Data are provided as a Source Data file.

Aponte et al. (2023) identified elemental sulfur ($S_8$) in the methanol extracts from Ryugu samples A0106 and C0107[14]. Interestingly, trithionic acid ($H_2S_3O_6$), the least stable polythionic acid whose end products are elemental sulfur and sulfates, at the surface of Ryugu (i.e., sample A0106) is three orders of magnitude less abundant than in sample C0107 (Fig. 4). Surface irradiation might efficiently convert thiosulfates to stable allotropes of $S_n$.

Our analysis of samples from Ryugu has revealed that the abundant soluble sulfur-containing compounds have undergone diverse chemical evolution. The careful curation and lack of opportunities for uncontrolled sample exposure to terrestrial weathering[19,40] indicate that these compounds are indigenous to Ryugu. Our findings suggest that such transformation reactions could have directly affected the chemical behavior (i.e., the hydrophilic, hydrophobic, and amphiphilic properties) of organic matter. The SOM content of A0106 is less than that of the CM Murchison and comparable to the SOM contents of the unheated CI Ivuna and Orgueil[4,10,24]. There is a growing interest in the role of minerals and metals in the co-evolution of organic and inorganic matter[11,41], and the functionality and abundance of organic matter are specific to mineralogical lithologies[8]. Furthermore, a sample of the carbonaceous asteroid (101955) Bennu is scheduled to be returned in 2023 by the OSIRIS-REx mission. During investigation of Bennu, large veins of calcite, dolomite–breunnerite, and magnesite, presumably formed by aqueous alteration processes, were detected[42]. Future work is expected to verify whether differences in water content and thermal history result in a variety of soluble amphibolic compounds being produced during S-bearing molecular evolution.

## Methods
### Samples and sequential chemical extraction
Samples A0106 and C0107 from Ryugu, and material from the Orgueil, Tarda, Aguas Zarcas, and Jbilet Winselwan meteorites, were studied.

The samples were collected at the first touchdown site (A0106) and the second touchdown site (C0107) on the asteroid Ryugu[1,2]; C0107 contains subsurface samples from the artificially made impact crater[43].

Sample weights used for chemical extraction treatments #7-1 to #10 in Supplementary Fig. 1B were 17.15 mg of A0106, 17.36 mg of C0107, and 17.91 mg of Orgueil meteorite (CI type, from National Museum of Denmark). A sequential extraction was performed on these samples using 600 μL of each of the solvents in the order hot $H_2O$ (#7-1), dichloromethane and methanol (#8), HCOOH (#9), and HCl (#10). The samples sealed in a vacuum were first extracted with #7-1 hot $H_2O$, weighed immediately after vial opening, and reacted with $N_2$ gas-purged ultrapure water (Tamapure AA100, Tama Chemical) for 20 h at 105 °C in a flame-sealed ampoule with the headspace also purged with $N_2$ gas. The reaction tube was centrifuged at 13,150 x *g* for 8 min, opened, and the supernatant was recovered. Dichloromethane (DCM, PCB analysis grade, FUJIFILM Wako Pure Chemical Corporation) and methanol (MeOH, QTofMS analysis grade, Fujifilm Wako pure chemical corporation) were then mixed at a volume ratio of 1:1. The DCM/MeOH extraction (#8) was carried out in an ultrasonic bath (Branson, CPX1800-J) at room temperature for 15 min, and the supernatant was recovered after centrifugation at 9660 x *g* for 5 min. Next, >99% HCOOH (#9, Fujifilm Wako pure chemical corporation) was reacted overnight at room temperature. Finally, a 15-min sonication with 20% HCl (#10, Tamapure AA100, Tama Chemical) at room temperature was performed to complete the extraction of soluble substances. The #9 and #10 fractions were centrifuged under the same conditions as ultrapure water.

Note that the amounts of major cations and anions in each fraction are divided by the initial weights of the starting solid materials used for sequential leaching (in μmol/g, Figs. 1, 3, 5). The percentages of clay mineral and carbonate standards dissolved in the HCOOH and HCl are shown in Supplementary Table 5. Samples from Tarda

(15.76 mg), Aguas Zarcas (15.51 mg), and Jbilet Winselwan (14.90 mg) were also subjected to the same extraction experiments for comparison, but only the HCOOH and HCl fractions could be used in this study because they were preferentially used for the analysis of other soluble organic matter. The results for these meteorites are also shown in Supplementary Table 6. This method was previously applied to carbonaceous chondrites such as Murchison as a rehearsal analysis for the Hayabusa2 project, and the concentrations of dissolved constituents in the HCOOH and HCl fractions have been reported[44].

The other chemical extraction step using organic solvents and ultrapure water is shown in Supplementary Fig. 1B. Sample weights used for #2 to #5 were 17.15 mg of A0106, 17.36 mg of C0107, and 17.56 mg of Orgueil. These samples were subjected to sequential extraction with organic solvents in the order hexane (#2), dichloromethane (#3), and methanol (#4). At each step, the extract was sonicated for 15 min, after which the supernatant was recovered. At the end of the organic solvent extraction, the fractions were extracted with ultrapure water (#5) at room temperature, but only the Orgueil results could be used for this study.

## Ion chromatography

After the extraction processes, the anion and cation concentrations of each fraction were measured by ion chromatography (IC), using the Metrohm 930 Compact IC Flex system (Metrohm AG, Herisau, Switzerland). For cations, the samples were eluted through a Metrohm Metrosep C6-250/4.0 column with 8 mM ultrapure $HNO_3$ (TAMAPURE AA-100, Tama Chemical, Kawasaki, Japan) at a flow rate of $0.9 \, mL \cdot min^{-1}$. Anions were measured with a Metrohm Metrosep A Supp4-250/4.0 column with a chemical suppressor module. The mobile phase consisted of a mixture of 1.8 mM $Na_2CO_3$ and 1.7 mM $NaHCO_3$ (Kanto Chemical, Tokyo, Japan) at a flow rate of $0.9 \, mL \cdot min^{-1}$. A chemical suppressor module (Metrohm MSM) was used to decrease the background conductivity of the eluent and to transform the analytes into free anions. The column temperature was set at 35 °C throughout the analysis. Detection of cations and anions was accomplished by measuring electrical conductivity.

## Ion chromatography/Mass spectrometry

The distribution of anions in the Ryugu extracts was analyzed by using a Dionex ICS-6000 IC system (Thermo Fisher Scientific Inc., Waltham, USA) equipped with an Orbitrap Exploris 480 mass spectrometer (Thermo Fisher Scientific Inc., Waltham, USA). For the IC separation, a Dionex IonPac® AS11-HC analytical column (2 × 250 mm, Thermo Fisher Scientific Inc., Waltham, USA) with a guard column was used at 35 °C. The mobile phase was KOH at a constant flow rate of $0.25 \, mL \cdot min^{-1}$. The KOH gradient program was 1.0 mM KOH from 0 to 1 min, which was increased to 50.0 mM KOH from 1 to 40 min and held there until 64 min, after which it was decreased to 1.0 mM KOH from 64 to 65 min and held there until 75 min. A conductivity detector combined with an AERS suppressor (Thermo Fisher Scientific Inc., Waltham, USA) was utilized to cross-check the detection of anions by the subsequent mass spectrometry.

The Orbitrap mass spectrometer was equipped with an electrospray ionization (ESI) source and operated in negative ion mode. The flow rates of nitrogen gas for desolvation were set to 40 arbitrary units (Arb) of the sheath gas, 5 Arb for the auxiliary gas, and 0 Arb for the sweep gas. The ion transfer capillary temperature and ESI spray voltage were set to 320 °C and 2.5 kV, respectively. Full scan mass spectra were acquired over a mass range of $m/z$ 50 to 750 with a mass resolution of 120,000 (at full-width-half-maximum for $m/z$ 200). Most ions were detected in the deprotonated form, $[M-H]^-$. The full scan measurements exhibited a general mass accuracy of less than 1 ppm, defined as [(measured $m/z$) − (calculated $m/z$)]/ (calculated $m/z$) × $10^6$ (ppm). An exclusion list composed of the two largest background peaks, $m/z$ 112.9856 (corresponding to $CF_3COO^-$) and $m/z$ 68.9958 (corresponding to $CF_3^-$) was implemented with a mass width of ±10 ppm to decrease the background signal.

## Inductively coupled plasma mass spectrometry

Trace-element concentrations were measured by quadrupole inductively coupled plasma mass spectrometry (ICP-MS, iCAP Qc, Thermo Fisher Scientific Inc., Waltham, USA). A 0.3 M $HNO_3$ solution was added to each vial to dilute the samples. The $HNO_3$ used in this study was a commercially supplied high-purity TAMAPURE AA-100 reagent (Tama Chemical, Kawasaki, Japan). We added internal standards (Be, Sc, Y, and In) to the $HNO_3$ to correct for the instrumental drift.

For the Mg isotope analysis, each extract was dried down, and then re-dissolved in 8 mM $HNO_3$. Samples were purified by an IC Metrohm 930 Compact IC Flex system coupled to an Agilent 1260 Infinity II Bio-Inert analytical-scale fraction collector system (Agilent Technologies, Santa Clara, USA) set in a class-1000 clean hood[45,46]. For complete separation of cations, the samples were eluted through a Metrohm Metrosep C6-250/4.0 column with 8 mM ultrapure $HNO_3$ at a flow rate of $0.9 \, mL \cdot min^{-1}$. Magnesium isotope ratios were measured by a multiple collector (MC) ICP-MS Neptune plus (Thermo Fisher Scientific Inc., Waltham, USA). We performed Mg isotope analysis with a high-sensitivity X-skimmer cone. Sample solutions were introduced with a PFA nebulizer (MicroFlow, ~50 μL·min$^{-1}$, ESI, Omaha, USA) attached to a quartz dual-cyclonic spray chamber in free aspiration mode. The beam intensity for the 100 ppb solutions was approximately 5.0 V for $^{24}Mg$. After initial uptake of the solutions, a single analysis consisted of 40 cycles with an integration time of 4 s per cycle. The background signal intensities were measured with a 0.3 M ultrapure $HNO_3$ solution for 1 cycle with an integration time of 30 s per cycle.

The isotopic data are expressed as per mil (‰) deviations relative to the DSM-3 standard. The Mg isotope ratio was defined as follows:

$$\delta^{26}Mg = \{(^{26}Mg/^{24}Mg)_{sample}/(^{26}Mg/^{24}Mg)_{DSM-3} - 1\} \times 1000 \quad (6)$$

## pH measurement

After recovering the supernatant by centrifugation during the sequential extraction[4], we performed pH measurements of the hot water fraction (#7-1 for A0106 and C0107) at 24.6 °C (within ±0.1 °C at ambient atmosphere) by using a LAQUA F-73 instrument (HORIBA Advanced Techno Co., Ltd., Kyoto, Japan) with a pH electrode (model 0040-10D), with an instrument repeatability of within ±0.001 of the pH value. In this pH measurement process, the small recovered supernatant water fraction (<10 μL) was measured without any dilution. Prior to the measurement, a three-point calibration was performed with phosphate standard solutions of pH 4.005 and 6.865 (both at ~25 °C) and a tetraborate standard solution of pH 9.18 (Kanto Chemical Co. Inc) (Supplementary Fig. 6).

## Data availability

Source data for figures are provided with the paper as a Source Data file and available from the corresponding author. The Hayabusa2 project is releasing raw data on the properties of the asteroid Ryugu from the Hayabusa2 Science Data Archives (DARTS, https://www.darts. isas.jaxa.jp/planet/project/hayabusa2/). We declare that all these database publications are compliant with ISAS data policies (https:// www.isas.jaxa.jp/en/researchers/data-policy/). Source data are provided with this paper.

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

## Acknowledgements

The Hayabusa2 project was led by ISAS (Institute of Space and Astronautical Science)/JAXA (Japan Aerospace Exploration Agency) in collaboration with DLR (German Space Center) and CNES (French Space Center), and supported by NASA (National Aeronautics and Space Administration) and ASA (Australian Space Agency). We thank the members of the Astromaterials Science Research Group (ASRG) at ISAS, and the Hayabusa2 curation team for conducting the sampling and quality control management. We express our deep appreciation for the constructive and insightful comments from Dr. E. Quirico. We also thank Y. Nakanishi of Thermo Fisher Scientific Inc. for support of the chemical assessment and the molecular-specific identification by mass spectrometry; Y. Yoshikawa of JAMSTEC for laboratory assistance; Mr. Kumazoe of Kyushu University for solvent extraction of the Tarda, Aguas Zarcas, and Jbilet Winselwan meteorites; Y. Kobayashi of Metrohm Japan for technical support with the ion chromatography; and Dr. Y Tamenori for advice on the chemical species of sulfur. Preliminary reports based on the current results were presented at the Lunar and Planetary Science Conference (LPSC) in 2020 and 2022. This research was partly supported by the Japan Society for the Promotion of Science (JSPS) under KAKENHI grant numbers 21H01204 (TYoshimura), 21H04501&21H05414 (YO), 21J00504 (TK), 21KK0062 (YT), and 20H00202 (HN). JPD and JCA are grateful to NASA for support of the Consortium for Hayabusa2 Analysis of Organic Solubles.

## Author contributions

TYoshimura, YTakano, H.N., and J.P.D. conceived the study. H.N. and YTakano conducted the sequential extraction and distributed the SOM samples. TYoshimura conducted IC analysis. D.A. and TYoshimura conducted the ICP analysis and the evaluation of the Mg isotopic composition. H.N., Y.O., T.K., MYamaguchi, and TSuzuki conducted the Orbitrap analysis. YTakano, TYoshikawa, and SatoruTanaka performed the small-scale aqueous analysis of pH with authentic standards. N.O.O. and N.O. performed small-scale elemental analysis of sulfur and isotopic composition, and provided interpretation of the sulfur chemistry. M.H., H.M., and E.P. supported the work flow on sequential extraction of reference sample processes with H.N. and YTakano. TYokoyama, HYurimoto, and STachibana provided the interpretation of asteroidal chemistry. H.N., YTakano, and J.P.D. designed the implementation of the SOM scheme prior to the initial analysis (until ~31 May 2022). P.S.-K., N.H., J.A., S.S., HYurimoto, TNakamura, TNoguchi, R.O., HYabuta, K.S., TYada, M.N., A.N., A.M., K.Y., M.A., T.O., T.U., MYoshikawa, TSaiki, SatoshiTanaka, F.T., S.N., S.W., YTsuda, STachibana, TYoshimura, H.N., T.K., D.A., N.O.O., Y.O., J.P.D., TYoshikawa, SatoruTanaka, N.O., M.H., H.M., E.P., MYamaguchi, TSuzuki, TYokoyama, YTakano, SOM contributed to the data analysis and manuscript revision, and read and approved the submitted version.

## Competing interests

The authors declare no competing interests.

## Additional information

¹Biogeochemistry Research Center (BGC), Japan Agency for Marine-Earth Science and Technology (JAMSTEC), Natsushima 2-15, Yokosuka, Kanagawa 237-0061, Japan. ²Department of Earth and Planetary Sciences, Kyushu University, 744 Motooka, Nishi-ku, Fukuoka 819-0395, Japan. ³Geological Survey of Japan (GSJ), National Institute of Advanced Industrial Science and Technology (AIST), 1-1-1 Higashi, Tsukuba, Ibaraki 305-8567, Japan. ⁴Helmholtz Zentrum München, Analytical BioGeoChemistry, Ingolstaedter Landstrasse 1, 85764 Neuherberg, Germany. ⁵Technische Universität München, Analytische Lebensmittel Chemie, Maximus-von-Forum 2, 85354 Freising, Germany. ⁶Institute of Low Temperature Science (ILTS), Hokkaido University, N19W8 Kita-ku, Sapporo 060-0189, Japan. ⁷Solar System Exploration Division, NASA Goddard Space Flight Center, Greenbelt, MD 20771, USA. ⁸HORIBA Advanced Techno, Co., Ltd.,

Kisshoin, Minami-ku, Kyoto 601-8510, Japan. [9]HORIBA Techno Service Co., Ltd. Kisshoin, Minami-ku, Kyoto 601-8510, Japan. [10]Department of Earth and Planetary Sciences, Nagoya University, Nagoya 464-8601, Japan. [11]Thermo Fisher Scientific Inc., 3-9 Moriyacho, Kanagawa-ku, Yokohama-shi, Kanagawa 221-0022, Japan. [12]Department of Earth and Planetary Sciences, Tokyo Institute of Technology, Ookayama, Meguro, Tokyo 152-8551, Japan. [13]Creative Research Institution (CRIS), Hokkaido University, Sapporo, Hokkaido 001-0021, Japan. [14]Department of Earth Science, Tohoku University, Sendai 980-8678, Japan. [15]Department of Earth and Planetary Sciences, Kyoto University, Kyoto 606-8502, Japan. [16]Earth and Planetary Systems Science Program, Hiroshima University, Higashi Hiroshima 739-8526, Japan. [17]Institute of Space and Astro-nautical Science, Japan Aerospace Exploration Agency (ISAS/JAXA), Sagamihara, Kanagawa 229-8510, Japan. [18]Kanagawa Institute of Technology, Atsugi 243-0292, Japan. [19]UTokyo Organization for Planetary and Space Science (UTOPS), University of Tokyo, Bunkyo-ku, Tokyo 113-0033, Japan. [40]These authors contributed equally: Toshihiro Yoshimura, Yoshinori Takano.
✉e-mail: yoshimurat@jamstec.go.jp

## Hayabusa2-initial-analysis SOM team

Hiroshi Naraoka[2], Yoshinori Takano [1,40], Jason P. Dworkin [7], Kenji Hamase[20], Aogu Furusho[20], Minako Hashiguchi [10], Kazuhiko Fukushima[21], Dan Aoki[21], José C. Aponte [7], Eric T. Parker[7], Daniel P. Glavin[7], Hannah L. McLain[7,22,23], Jamie E. Elsila[7], Heather V. Graham[7], John M. Eiler[24], Philippe Schmitt-Kopplin [4,5], Norbert Hertkorn [4], Alexander Ruf[25,26,27], Francois-Regis Orthous-Daunay[28], Cédric Wolters[28], Junko Isa[29,30], Véronique Vuitton[28], Roland Thissen[31], Nanako O. Ogawa [1], Saburo Sakai[1], Toshihiro Yoshimura [1,40]✉, Toshiki Koga [1], Haruna Sugahara[17], Naohiko Ohkouchi[1], Hajime Mita[32], Yoshihiro Furukawa[33], Yasuhiro Oba [6], Yoshito Chikaraishi[6], Takaaki Yoshikawa[8], Satoru Tanaka[9], Mayu Morita[34], Morihiko Onose[34], Daisuke Araoka [3], Fumie Kabashima[35], Kosuke Fujishima[29], Hajime Sato[36], Kazunori Sasaki[36,37], Kuniyuki Kano[38], Shin-ichiro M. Nomura[39], Junken Aoki[38], Tomoya Yamazaki[6] & Yuki Kimura[6]

[20]Graduate School of Pharmaceutical Sciences, Kyushu University, Fukuoka 812-8582, Japan. [21]Graduate School of Bioagricultural Sciences, Nagoya University, Nagoya 464-8601, Japan. [22]Center for Research and Exploration in Space Science and Technology, NASA Goddard Space Flight Center, Greenbelt, MD 20771, USA. [23]Department of Physics, The Catholic University of America, Washington, DC 20064, USA. [24]Division of Geological and Planetary Sciences, California Institute of Technology, Pasadena, CA 91125, USA. [25]Université Aix-Marseille, CNRS, Laboratoire de Physique des Interactions Ioniques et Moléculaires, Marseille 13397, France. [26]Department of Chemistry and Pharmacy, Ludwig-Maximilians-University, Munich 81377, Germany. [27]Excellence Cluster ORIGINS, Garching 85748, Germany. [28]Université Grenoble Alpes, Centre National de la Recherche Scientifique (CNRS), Centre National d'Etudes Spatiales, L'Institut de Planétologie et d'Astrophysique de Grenoble, Grenoble 38000, France. [29]Earth-Life Science Institute (ELSI), Tokyo Institute of Technology, Tokyo 152-8550, Japan. [30]Planetary Exploration Research Center, Chiba Institute of Technology, Narashino 275-0016, Japan. [31]Université Paris-Saclay, CNRS, Institut de Chimie Physique, Orsay 91405, France. [32]Department of Life, Environment and Material Science, Fukuoka Institute of Technology, Fukuoka 811-0295, Japan. [33]Department of Earth Science, Tohoku University, Sendai 980-8678, Japan. [34]HORIBA Techno Service Co., Ltd., Kyoto 601-8305, Japan. [35]LECO Japan Corp., Tokyo 105-0014, Japan. [36]Institute for Advanced Biosciences (IAB), Keio University, Kakuganji, Tsuruoka, Yamagata 997-0052, Japan. [37]Human Metabolome Technologies (HMT) Inc., Kakuganji, Tsuruoka, Yamagata 997-0052, Japan. [38]Department of Health Chemistry, Graduate School of Pharmaceutical Sciences, The University of Tokyo, Hongo, Tokyo 113-0033, Japan. [39]Department of Robotics, Graduate school of Engineering, Tohoku University, Sendai, Miyagi 980-8579, Japan.

