## [Peer Review File NEW · Nature Communications]

Chemical evolution of primordial salts and organic sulfur molecules in the asteroid 162173 RyuguReviewer #1 (Remarks to the Author):

The paper "Chemical evolution of primordial salts and organic sulfur molecules in the asteroid (162173) Ryugu" by Toshihiro Yoshimura et al. deals with the analysis of two samples from the Hayabusha2 mission looking at SOM's, especially at their salt fraction. These are very new and significant data. The Ryugu samples show clear aqueous alterations, leading to relatively high abundances of CHONS material. What makes especially interesting reading is the part about sulfur chemistry in these samples. A comparison with carbonaceous meteorites leads to the conclusion, that meteoritic material shows clear signs of alterations during their terrestrial journey. The paper is well written and very comprehensive. The analysis is described in detail and the results are sound.

The only comment I have is that the result section is rather short and the discussion section rather long. The discussion section is a mix between results and actual discussion about the meaning of the results. The paper would be better readable if the part of the discussion section, which contains results, is moved to results.

Reviewer #2 (Remarks to the Author):

This manuscript by T. Yoshimura reports the analysis of salts and organic sulfur molecules in Ryugu samples collected at the surface of the Ryugu asteroid. Both collectors A and C were analyzed. Four types of anions have been identified as SO_4^{2-} , Cl^- , S_2O_3^- and NO_3^- . Na^+ is the most abundant cation. Along with salts, several sulfur-bearing organic molecules were identified by HighResolution Mass Spectrometry.

Ryugu samples are linked to the very rare group of CI chondrites, which are known to be extremely sensitive to Earth conditions (see Pisani, 1864; Gounelle and Zolensky, 2001). Ryugu samples offer unique opportunities to characterize fragile semi-volatile compounds like salts. This publication reports unprecedented results, which are of primary importance for cosmochemistry.

I recommend the publication of this manuscript, please see below my comments.

Kind Regards,
Eric Quirico

L69-71: NH_4^+ is not listed in the abstract, but it is mentioned on pages 30 and 31.

L133-137: The "missing sulfur problem" concerns sulfur in molecular clouds and dense cores. The abundance of sulfur in small molecules in the gas phase does not reach the cosmic value, which means that a sulfur reservoir is hidden somewhere. There is no consensus on this issue, but it seems that sulfur could be present at the surface of grains as organic sulfur molecules (e.g. Laas et al., 2019) or elemental sulfur produced by GCR radiolysis (Shingledecker et al., 2020). However, there is no missing sulfur in chondrites, and sulfur abundance (normalized to Si) is similar to that of the Sun.

L154-155: Four extractions were run: hot ultra-pure water, Me-OH, and HCl. Regarding phyllosilicates, do you mean that ions were trapped in the interlayer space of saponite, and then expelled by Cl^- exchange?

Another point is that some sulfides are soluble in HCl and release sulfur (this is clearly observed during the HCl sequence of Insoluble Organic Matter extraction, the supernatant turns to yellow/green). Can you warrant this sulfur does not interfere with your other analyzes?

L214-215: I think this issue was fixed already by Gounelle and Zolensky (2001). In Orgueil, a fraction of sulfates is terrestrial, but not all sulfates.

L251: I don't understand how this equation is introduced. Spatolisano+2021 is not relevant to asteroidal conditions, it is an Engineering publication. At L254, this equation is even modified, by replacing H₂S with (Fe, Ni)S. Why remove H₂O and keep SO₂? What is the justification of this equation? It seems it comes out of the blue.

L256 Zolotov (2012) is cited several times. I think it would be helpful to the reader to add a text section that summarizes how the results of this very comprehensive thermodynamical modelling are consistent with the manuscript's new findings.

L310-311: As pointed out before, the "missing sulfur problem" concerns molecular clouds and dense cores, and cannot be solved by analyzing Ryugu grains.

L316: Surface irradiation by what? There is no reference here, and in any case, the electronic and nuclear doses should be estimated. Experimental studies are also necessary to conclude about the conditions that lead to the formation of S allotropes.

L327-328: A CI chondrite is very different from comet 67P/CG, which has never experienced intense aqueous alteration. One salt has been detected (NH₄⁺; HS⁻) by both ROSINA and VIRTIS instruments. In that case, it is likely a solid-state thermally activated reaction between H₂S and NH₃. To say this is quite disconnected from Ryugu samples.

Reviewer #3 (Remarks to the Author):

The authors report the results of anion and cation analysis of Ryugu samples and demonstrate the abundance of soluble sulfur-bearing compounds. Based on the results, the authors propose transformation mechanisms that could explain the formation of various sulfur-bearing species in Ryugu. The results and proposed mechanism are important for understanding sulfur evolution in the early solar system which is still poorly constrained.

The authors performed sequential extractions; however, the methodology is poorly explained which does not allow the methods to be reproduced and to evaluate the data and efficiency of the carbonate/phyllsilicate extraction steps. In my opinion, some of the data interpretation and the proposed transformation mechanisms of sulfur-bearing compounds need to be addressed prior to being considered for publication. My comments are summarized below.

Main comments:

1. Extraction procedure

Sequential extraction procedure steps are poorly described which makes it hard to evaluate the data and efficiency of carbonate/phyllsilicate extraction steps. I recommend the following information be added and questions be addressed

#7-1 Please add water volume, temperature, and extraction duration. Was O₂-free water used for this extraction to avoid oxidation of sulfides? Was extraction performed in an inert atmosphere (ex., N₂, Ar)?

#7-2 Please add HCl volume, concentration, and extraction duration

#8 Please add DCM/MeOH ratio, extraction duration, volume

#9 Please add HCOOH concentration, volume, duration

#10 Please add HCl concentration, volume, duration

The references must be provided to support that #9 HCOOH and #10 HCl extractions are selective extractions for carbonates and phyllosilicates. Are carbonates and phyllosilicates completely dissolved? Are other phases extracted by HCOOH and HCl? The presence of both Fe and Ni in the extracted solutions (Supplementary Table 1) is indicative that sulfides were also dissolved.

2. Acidic pH of the hot water extraction (l. 177-180)

Why pH was acidic at the end of hot water extraction? The presence of strong base cations (Na, Ca, Mg), strong acid anions (SO₄, Cl, NO₃), and weak acid anions (S₂O₃) could lead to neutral or alkaline pH. Is it possible that sulfide oxidative dissolution occurred during this extraction step resulting in a pH decrease and release of SO₄²⁻ and S₂O₃²⁻?

3. Origin of sulfate/thiosulfate/chloride detected in the hot water extraction

The authors discuss that soluble inorganic salts, including chlorides and sulfates, were not detected (or were present in trace amounts) in Ryugu samples (starting line 197), and then propose that organosulfur-bearing anions are counterions to Na. However, the manuscript lacks a clear discussion of where sulfate/thiosulfate/chloride were released from during the hot water extraction. In the case of sulfate/thiosulfate, where they present as adsorbed species, originated from organic sulfur-bearing species or from oxidizing sulfides?

4. Transformation mechanism of sulfur-containing species

The alteration model of Nakamura et al., 2022 shows that aqueous alteration of Ryugu resulted in alkaline conditions, however, the authors indicate that acidic conditions are necessary to stabilize polythionates. How did acidic conditions develop? Can the authors speculate about the timing of the process? Did it happen before or after the aqueous alteration? What water-to-rock ratio is necessary for reactions 2-4 to occur?

Other comments:

1. Online methods/ Ion chromatography: what is exchangeable carbonate?

2. Fig 1b caption: Please define Exchangeable ions. What minerals are they associated with?

"We measured evaporitic salts (via #7-1 hot water extraction, see IDs in Extended Data Fig. 2 and Naraoka+2023)"

Should it be Extended Data Fig. 1b?

3. l. 177-178 "The pH of the salt fraction at 24.6 °C (#7-1 in Supplementary Fig. 1) was weakly acidic..."

why is there ref to this figure as no pH data is shown there?

4. Fig. 2 Please references where Ca, Mg, Na, and K were taken from for Orgueit, Tarda, Aguas Zarcas, and Jbilet Winselwan. If these samples were analyzed in this work, please add the data to Supplementary Table 1

"cosmic abundance (stars) also plotted for reference" Please add references

5. Fig. 3 Dissolved carbonate is shown in (a) but it is unclear how it was detected. Was it measured with IC or was it a part of a different study? If the second, please add a reference

6. Suppl Table 2 " Note that dissolved inorganic carbon and silica were not measured"

HCO₃⁻ is reported in the Fig. 3a, please clarify

Replies to Reviewer's comments on the manuscript #NCOMMS- 23-08319-T

We appreciate the constructive comments on our manuscript (#NCOMMS-23-08319-T) entitled “**Chemical evolution of primordial salts and organic sulfur molecules in the asteroid (162173) Ryugu**” (by Yoshimura et al.). We carefully read all of the comments in the reviews and modified the manuscript based on your helpful feedback. The changes that we made based on the comments are shown in red text in the revised manuscript/supporting information. We have chosen the option of "Transparent peer review", to make public the discussion during the review process. We believe our point-by-point responses to each comment will clarify the entire context of the paper.

Reviewer #1 (Remarks to the Author):

The only comment I have is that the result section is rather short and the discussion section rather long. The discussion section is a mix between results and actual discussion about the meaning of the results. The paper would be better readable if the part of the discussion section, which contains results, is moved to results.

→ We thank the reviewer for their advice. We have reorganized the Results and Discussion sections. Please see the revised manuscript.

Reviewer #2 (Remarks to the Author):

L69-71: NH_4^+ is not listed in the abstract, but it is mentioned on pages 30 and 31.

→ We revised the abstract as suggested: “Anions and NH_4^+ were more abundant in the salt fraction than in the carbonate and phyllosilicate fractions, with molar concentrations in the following order: $\text{SO}_4^{2-} > \text{Cl}^- > \text{S}_2\text{O}_3^{2-} > \text{NO}_3^- > \text{NH}_4^+$ ”. As the reviewer pointed out, we also detected ammonia, but we included most of the information on NH_4^+ in the supplementary information, since the matter was not directly related to the discussion of sulfur compounds. Reviewer #1 also commented that we should include additional description of the results, so

we have added a description of the NH_4^+ to the Results section.

L133-137: The “missing sulfur problem” concerns sulfur in molecular clouds and dense cores. The abundance of sulfur in small molecules in the gas phase does not reach the cosmic value, which means that a sulfur reservoir is hidden somewhere. There is no consensus on this issue, but it seems that sulfur could be present at the surface of grains as organic sulfur molecules (e.g. Laas et al., 2019) or elemental sulfur produced by GCR radiolysis (Shingledecker et al., 2020). However, there is no missing sulfur in chondrites, and sulfur abundance (normalized to Si) is similar to that of the Sun.

→ We agree. Since there is still some uncertainty about this matter, we revised the text to tone down this point.

L310-311: As pointed out before, the “missing sulfur problem” concerns molecular clouds and dense cores, and cannot be solved by analyzing Ryugu grains.

→ These two lines (L136 and 340 of marked manuscript) were removed from the main text.

L316: Surface irradiation by what? There is no reference here, and in any case, the electronic and nuclear doses should be estimated. Experimental studies are also necessary to conclude about the conditions that lead to the formation of S allotropes.

→ According to Okazaki (2023, Science), the galactic cosmic ray (GCR) irradiation period of the Ryugu samples is 5 million years based on the result of GCR-produced noble gases. Nishiizumi et al. (2022, LPSC, Reference below) also give irradiation ages determined by cosmogenic nuclides on the order of millions of years. For formation of S allotropes, we considered cosmic-ray radiation as proposed by Shingledecker et al. (2020, Astrophys. J.). Although we now have new data on the reaction pathways of sulfur species, we agree that there is a difference in the environment of formation, as the reviewer pointed out, and that additional verification is needed to suggest the existence of similar processes from the present data. We have revised lines 134 and 138 of the manuscript according to this comment, and added that future experimental verification is required.

[Reference]

Nishiizumi, K. et al. Exposure conditions of samples collected on Ryugu's two touchdown sites determined by cosmogenic nuclides ^{10}Be and ^{26}Al . Lunar and Planetary Science

Conference (LPSC), #1777 (2022).

L154-155: Four extractions were run: hot ultra-pure water, Me-OH, and HCl. Regarding phyllosilicates, do you mean that ions were trapped in the interlayer space of saponite, and then expelled by Cl⁻ exchange?

Another point is that some sulfides are soluble in HCl and release sulfur (this is clearly observed during the HCl sequence of Insoluble Organic Matter extraction, the supernatant turns to yellow/green). Can you warrant this sulfur does not interfere with your other analyzes?

→ In the reaction with HCl, a color change of the supernatant was observed (see the photograph below). We reported various S-bearing compounds from the #7-1 hot H₂O fraction. In contrast, very small amounts of S-bearing compounds were obtained from the #9 HCOOH and #10 HCl fractions, as confirmed by our Orbitrap-MS analysis. The Fe concentrations of A0106 and C0107 are 0.16 and 0.37 mmol/g, respectively. If we assume that the S measured in the #7-1 fraction was all derived from FeS, and ignore magnetite and phyllosilicates, the amount of S released was about 1/600 of the total amount of SO₄²⁻ and 1/400 of the total amount of S₂O₃²⁻ in the #7-1 fractions of A0106 and C0107. Thus, the effect of sulfide dissolution on the composition of dissolved species in the solution was small. Sulfide dissolution mainly occurred in the #10 HCl and #9 HCOOH fractions: ~1000 to 8000 times more Fe was detected in the HCOOH and HCl fractions than in the #7-1 hot H₂O fraction. In addition, the calculated Mg# (Mg/Mg+Fe) for the #7-1 fractions of Ryugu is ~82, which is within the typical range of values for Ryugu phyllosilicates (mainly 75 to 90, Nakamura et al., 2023, Science). Iron may be derived from secondary minerals other than FeS, in which case the effect of iron sulfide dissolution on sulfur would be smaller.

The reactions of HCl and HCOOH with the most common secondary minerals were also tested as part of method verification. Layered silicate minerals with interlayer water such as saponite are especially soluble in acids. We checked to what extent saponite, montmorillonite, pyrophyllite, dickite, and kaolinite published by the Japan Clay Science Society (JCSS) became dissolved in the acid reagents and experimental conditions used in this study. Saponite is the most abundant mixed-layer mineral at Ryugu (Yokoyama et al., 2023, Science; Nakamura et al., 2023, Science). Saponite is particularly soluble in hydrochloric acid: almost 100% of the standard saponite dissolved in HCl (see the table in our response to Reviewer #3). Therefore, it should be possible to decompose and extract the components adsorbed and structurally incorporated into the saponite, rather than only extracting the adsorbed ions of saponite. We have added this information as a supplement so that the experiment can be repeated in future studies.

L214-215: I think this issue was fixed already by Gounelle and Zolensky (2001). In Orgueil, a fraction of sulfates is terrestrial, but not all sulfates.

- We have revised the text to emphasize that Mg sulfate is precipitated by terrestrial weathering. The detailed history of the samples studied by Gounelle and Zolensky (2001) suggests that the Mg sulfate is terrestrial, and the sulfate $\Delta^{17}\text{O}$ and $^{87}\text{Sr}/^{86}\text{Sr}$ values also suggest that it was precipitated on the Earth (references in Gounelle and Zolensky, 2001). However, the source of Mg may be a new finding. We have revised the text to clarify that the novelty of the result lies in the fact that the $d^{26}\text{Mg}$ of the hot H_2O fraction containing Mg sulfate is similar to that of the formic acid fraction, indicating that the Mg was derived from dolomite.

L251: I don't understand how this equation is introduced. Spatolisano+2021 is not relevant to asteroidal conditions, it is an Engineering publication. At L254, this equation is even modified, by replacing H_2S with $(\text{Fe}, \text{Ni})\text{S}$. Why remove H_2O and keep SO_2 ? What is the justification of this equation? It seems it comes out of the blue.

- As pointed out by the reviewer, there is no doubt that a water-mediated reaction occurred, so we have deleted Equation 2. Because there are few good references that specify this reaction step, we cited this ion chromatography analysis paper. The lack of reference papers is probably due to analytical complications, because many sulfur species are intermediate products in the oxidative environment on Earth. It was difficult to determine whether liquid-phase or solid-phase reactions were predominant,

so we proposed both. The coexistence of dissolved sulfur species and dissolved organic matter in the fluid inclusions of pyrrhotite has been reported by Nakamura et al. (2023, Science).

L256 Zolotov (2012) is cited several times. I think it would be helpful to the reader to add a text section that summarizes how the results of this very comprehensive thermodynamical modelling are consistent with the manuscript's new findings.

→ We added the following summary of the chemical modeling results.

“Chemical equilibrium modeling of aqueous alteration of the Ryugu parent body, with mixing of rocks, water containing CO₂ and HCl, and organic matter, yields low Water/Rock ratios (W/R, ranging from 0.06–0.1 for least-altered to 0.2–0.9 for extensively altered lithologies) and high Na concentrations in both fluids and the secondary mineral saponite (Nakamura+2023). Previous comprehensive thermodynamical modeling has demonstrated that, under low W/R conditions, neutralization of the initial HCl-containing acidic solution results in a Na-rich alkaline fluid in which Na-containing secondary minerals such as saponite can stably exist (Zolotov, 2012). The Ryugu results suggest the existence of Mg–Na–Cl-rich solutions in the early stages of aqueous alteration, which evolved into more reductive, Na–Cl alkaline brines that coexisted with H₂-rich gas phases (Nakamura+2023). The solutes with high solubility extracted by our hot H₂O method also yield a Na-rich composition, consistent with the chemical modeling.”

L327-328: A CI chondrite is very different from comet 67P/CG, which has never experienced intense aqueous alteration. One salt has been detected (NH₄⁺; HS⁻) by both ROSINA and VIRTIS instruments. In that case, it is likely a solid-state thermally activated reaction between H₂S and NH₃. To say this is quite disconnected from Ryugu samples.

→ We agree with this comment and have revised the text.

Reviewer #3 (Remarks to the Author):

1. Extraction procedure

Sequential extraction procedure steps are poorly described which makes it hard to evaluate the data and efficiency of carbonate/phyllosilicate extraction steps. I recommend the following information be added and questions be addressed

#7-1 Please add water volume, temperature, and extraction duration. Was O₂-free water used for this extraction to avoid oxidation of sulfides? Was extraction performed in an inert atmosphere (ex., N₂, Ar)?

#7-2 Please add HCl volume, concentration, and extraction duration

→ Measurement of fO₂ is difficult, because only a small amount of solution (600 μL) was used under N₂-purged sealed conditions; thus, we would like to explain the mineral behaviors based on the elemental composition of the fractions. The sample was weighed immediately after the vials containing vacuumed Ryugu materials were opened. The reaction was started with ultrapure water with inert N₂ gas, and left to react for 20 h at 105°C. As mentioned in the response to reviewer #2's comment, the concentrations of Fe (which are considered to indicate sulfide dissolution) are very low (0.16 and 0.37 mmol/g for A0106 and C0107, respectively). In contrast, in the subsequent HCOOH and HCl extractions, approximately 1,000 to 8,000 times more iron was detected. Therefore, dissolution of sulfide was almost completely suppressed under inert conditions in the #7-1 hot H₂O extraction in which soluble organic matter (SOM) was detected in this study.

Details of the following experiment have been added to the Methods section. As pointed out by the reviewer, hydrolysis with HCl acid was also performed (fraction #7-2). This hydrolysis was done by dividing the sample in two after #7-1, then hydrolyzing one part at 100°C for 20 h. However, this fraction was not used in this study because it was preferentially used for other SOM analyses. There is also the possibility of confusion with the #10 HCl fraction, so we would like to omit explanation of fraction #7-2 from this study.

#8 Please add DCM/MeOH ratio, extraction duration, volume

#9 Please add HCOOH concentration, volume, duration

#10 Please add HCl concentration, volume, duration

→ We have included additional description of this sequential solvent extraction method, which follows Naraoka +Science (2023). “Sample weights used for chemical extraction treatments #7-1 to #10 in Extended Fig. 1B were 17.15 mg of A0106, 17.36 mg of C0107, and 17.91 mg of Orgueil meteorite (CI type, from National Museum of Denmark). A sequential extraction was performed on these samples using 600 μL of each of the solvents in the order hot H₂O (#7-1), dichloromethane and methanol (#8), HCOOH (#9), and HCl (#10). The samples sealed in a vacuum were first extracted with #7-1 hot H₂O, weighed immediately after vial opening, and reacted with N₂ gas-purged ultrapure water (Tama pure AA100, Tama Chemical) for 20 h at 105°C in a flame-sealed ampoule with the headspace also purged with

N₂ gas. The reaction tube was centrifuged at 14,000 rpm for 8 min, opened, and the supernatant was recovered. Dichloromethane (DCM, PCB analysis grade, FUJIFILM Wako Pure Chemical Corporation) and methanol (MeOH, QToFMS analysis grade, Fujifilm Wako pure chemical corporation) were then mixed at a volume ratio of 1:1. The DCM/MeOH extraction (#8) was carried out in an ultrasonic bath (Branson, CPX1800-J) at room temperature for 15 min, and the supernatant was recovered after centrifugation at 12,000 rpm for 5 min. Next, >99% HCOOH (#9, Fujifilm Wako pure chemical corporation) was reacted overnight at room temperature. Finally, a 15-min sonication with 20% HCl (#10, Tamapure AA100, Tama Chemical) at room temperature was performed to complete the extraction of soluble substances. The #9 and #10 fractions were centrifuged under the same conditions as ultrapure water.

Note that the amounts of major cations and anions in each fraction are divided by the initial weights of the starting solid materials used for sequential leaching (in $\mu\text{mol/g}$, Figs. 1, 3, 5). The percentages of clay mineral and carbonate standards dissolved in the HCOOH and HCl are shown in Supplementary Table 5. Samples from Tarda (15.76 mg), Aguas Zarcas (15.51 mg), and Jbilet Winselwan (14.90 mg) were also subjected to the same extraction experiments for comparison, but only the HCOOH and HCl fractions could be used in this study because they were preferentially used for the analysis of other soluble organic matter. The results for these meteorites are also shown in Supplementary Table 6. This method was previously applied to carbonaceous chondrites such as Murchison as a rehearsal analysis for the Hayabusa2 project, and the concentrations of dissolved constituents in the HCOOH and HCl fractions have been reported (Yoshimura+2020).

The other chemical extraction step using organic solvents and ultrapure water is shown in Extended Fig. 1B. Sample weights used for #2 to #5 were 17.15 mg of A0106, 17.36 mg of C0107, and 17.56 mg of Orgueil. These samples were subjected to sequential extraction with organic solvents in the order hexane (#2), dichloromethane (#3), and methanol (#4). At each step, the extract was sonicated for 15 min, after which the supernatant was recovered. At the end of the organic solvent extraction, the fractions were extracted with ultrapure water (#5) at room temperature, but only the Orgueil results could be used for this study.”

The references must be provided to support that #9 HCOOH and #10 HCl extractions are selective extractions for carbonates and phyllosilicates. Are carbonates and phyllosilicates completely dissolved? Are other phases extracted by HCOOH and HCl? The presence of both Fe and Ni in the extracted solutions (Supplementary Table 1) is indicative that sulfides were also dissolved.

→ In sequential leaching, mineral phases other than the main target minerals are also partly

dissolved according to their solubility. For this study, we verified that sufficient extraction was possible using standard samples of the main target minerals for dissolution, and have added this information to the Methods and Supplementary Information.

First, dolomite is the main carbonate mineral of Ryugu as well as other CI chondrites. We conducted an experiment in which the dolomite standard published by the National Institute of Advanced Industrial Science and Technology (AIST) was reacted with the formic acid used in this study under the same conditions. The dissolution rate was calculated from the Ca concentration in the HCOOH after the reaction. The dolomite yield was $103.3\% \pm 5.0\%$ (2SD), indicating that it was completely decomposed.

The solubility of clay minerals in HCOOH and HCl was verified by using JCSS reference materials. According to our experiment, almost all of the saponite became dissolved in 20% HCl (please see the table below). Other phyllosilicates, montmorillonite, dickite, kaolinite, and pyrophyllite were also tested in the same way. Dissolution of clay minerals was evaluated by measurement of the aluminum or magnesium concentration. The dissolution rate of saponite was $103.3\% \pm 2.2\%$ (2 SD). Although the concentrations varied slightly around 100%, the JCSS elemental reference values themselves differed by 8.0% (2SD) among three different laboratories (Miyawaki et al., 2010, Clay Science, in Japanese), so basically a 100% yield was achieved. The >99% HCOOH, which was the extraction fraction before #10 HCl, also dissolved about 10% of the saponite. The SOM extraction method of Naraoka et al. (2023, Science) used very concentrated formic acid, but the dissolution behavior of saponite varied with the acidity of the monocarboxylic acid according to the test using 6.0% (1M) CH₃COOH (see the table below). These results have also been added to the Supplementary Information to improve the reproducibility of the experiment protocol.

In addition, the amount of S released was about 1/600 and 1/400 of the total amount of SO₄²⁻ and S₂O₃²⁻, respectively, in the #7-1 hot H₂O fractions of A0106 and C0107. Sulfide dissolution mainly occurred in the #10 HCl and #9 HCOOH fractions. Approximately 1,000 to 8,000 times more Fe was detected in the HCOOH and HCl fractions than in the #7-1 hot H₂O fraction. In the BCR method, which is a commonly used stepwise extraction method for terrestrial soils, dissolution of sulfide requires a strong oxidation reaction with H₂O₂. HCl is a strongly acidic solution, although it basically does not have oxidizing power, and the largest dissolution of sulfide was observed in the #10 HCl fraction. We have added a description of these dissolution behaviors to the Results section.

[Reference]

Miyawaki, R., et al., 2010. Some Reference Data for the JCSS Clay Specimens. Journal of the Clay Science Society of Japan, 48, 158–198. (in Japanese with English abstract)

Reagent type	20% HCl		99.9% HCOOH		6% CH ₃ COOH	
Fraction number of Naraoka+2023	#10		#9		(not used in this study)	
Percentage dissolved	%	2SD	%	2SD	%	2SD
Clay						
Condition 1: 15 mg clay + 600 μL reagents						
JCSS-3501 Saponite	103.3%	2.3	10.6%	0.3	9.7%	0.1
JCSS-3101b Montmorillonite	3.1%	0.1	0.5%	<0.1	1.6%	<0.1
JCSS-2101 Pyrophyllite	0.6%	2.4	0.3%	<0.1	0.6%	<0.1
JCSS-1301 Dickite	0.9%	<0.1	0.2%	<0.1	0.5%	<0.1
JCSS-1101c Kaolinite	10.1%	0.2	1.6%	<0.1	2.6%	0.1
Condition 2: 1 mg clay + 600 μL reagents						
JCSS-3501 Saponite	97.7%	2.5	-	-	-	-
JCSS-3101b Montmorillonite	6.6%	0.8	-	-	-	-
Condition 3: 75 mg clay + 600 μL reagents						
JCSS-3501 Saponite	80.6%	2.3	-	-	-	-
JCSS-3101b Montmorillonite	5.3%	0.3	-	-	-	-
Carbonate						
AIST JDo-1 Dolomite	-	-	103.3%	2.6	-	-

Supplementary Table 5. The percentages of clay mineral and carbonate standards dissolved in the sequential extraction acids of Naraoka+2023a. The percentage of dissolution was calculated by measuring Al or Mg for clay minerals and Ca for carbonates. The ratio of solid to liquid phase of clay minerals is the same as the solid/liquid ratio of Naraoka+2023a, with 600 μL of reagent added for 15 mg of solids. As the abundance of carbonates in Ryugu is a few percent, the sample weight of dolomite experiment was reduced to 1.5 mg of sample.

2. Acidic pH of the hot water extraction (l. 177-180)

Why pH was acidic at the end of hot water extraction? The presence of strong base cations (Na, Ca, Mg), strong acid anions (SO₄, Cl, NO₃), and weak acid anions (S₂O₃) could lead to neutral or alkaline pH. Is it possible that sulfide oxidative dissolution occurred during this extraction step resulting in a pH decrease and release of SO₄²⁻ and S₂O₃²⁻?

→ The dissolution of sulfides is as described above and is small in the #7-1 fraction. Please also see our replies to Reviewer #2 and your comment 4 below.

3. Origin of sulfate/thiosulfate/chloride detected in the hot water extraction

The authors discuss that soluble inorganic salts, including chlorides and sulfates, were not detected (or were present in trace amounts) in Ryugu samples (starting line 197), and then propose that organosulfur-bearing anions are counterions to Na. However, the manuscript lacks a clear discussion of where sulfate/thiosulfate/chloride were released from during the hot water extraction. In the case of sulfate/thiosulfate, where they present as adsorbed species, originated

from organic sulfur-bearing species or from oxidizing sulfides?

→ In this report, we discussed the SOM of the hot H₂O fraction, from which most of the dissolved sulfur species were also eluted. The absence of inorganic sulfate salts has been mineralogically confirmed by Nakamura et al. (2023, Science), and the contribution from sulfides is only a few hundredths of that from dissolved S species, as discussed previously. As for adsorption on clay minerals, as the reviewer pointed out, saponite generally generates a negative charge by isomorphous substitution of Si⁴⁺ for Al³⁺ in the surface tetrahedral layer; thus, cation adsorption is basically predominant. Laboratory experiments on the pH dependence of anion exchange capacity (AEC) and cation exchange capacity (CEC) of smectite have shown that CEC increases from approximately 25 to 32 as pH rises from about 5 to 8, whereas AEC is very small and shows no change (Wada et al., 1981, see the figure below). The total ratio of cations and anions eluted by the hot H₂O extraction is approximately 3:1, but such a large amount of anions probably cannot be produced by desorption from saponite because of the ion balance. Therefore, it is likely that the anions are stored as functional groups of soluble organic matter or their salts. Information on these possibilities has been added to the revised manuscript.

Although it is difficult to specify the exact timing of molecular evolution, we assume that the main chemical composition of the soluble organic matter was determined during the latest or ongoing aqueous alteration processes. However, since we have not yet obtained any proof of this, we have avoided stating this (as also in Naraoka et al., 2023, Science). In recent years, there have been reports on the heterogeneity of the localization of Ryugu indigenous organic matter and the identification of chemical formulae in mineral grains (Hashiguchi et al. 2023, Earth Planets and Space; Schmitt-Kopplin et al. 2021, Hayabusa symposium). The latter paper is on Ryugu, but as the formation ages of carbonates, for example, are often constrained by Mn–Cr ages, the localization of sulfur-containing organic matter in specific minerals may allow us to determine the timing of molecular evolution in detail.

[Reference]

Schmitt-Kopplin, P. et al. Highest molecular diversity and structural complexity revealed with ultrahigh resolution mass spectrometry and nuclear magnetic resonance spectroscopy of Ryugu's samples. # Abstract S3-9, Hayabusa symposium (2021).

Redacted

Figure of the *pH* dependence of CEC and AEC of montmorillonites based on data of Wada 1981 (Shirozu, 2010, “*Introduction to clay mineralogy : fundamentals for clay science*” Asakura Publishing Co. Ltd.).

4. Transformation mechanism of sulfur-containing species

The alteration model of Nakamura et al., 2022 shows that aqueous alteration of Ryugu resulted in alkaline conditions, however, the authors indicate that acidic conditions are necessary to stabilize polythionates. How did acidic conditions develop? Can the authors speculate about the timing of the process? Did it happen before or after the aqueous alteration? What water-to-rock ratio is necessary for reactions 2-4 to occur?

→ As the reviewer pointed out, the *pH* calculated from the combination of secondary minerals is alkaline (Nakamura et al., 2023, Science), reflecting the *pH* of the fluids when episodes of secondary mineral precipitation occurred. Our measured *pH* does not include solutes from all secondary mineral phases precipitated from alkaline fluids such as carbonate and phyllosilicates, but only reflects the ionic balance of the organic and inorganic materials that are soluble in hot H₂O. Therefore, the measured *pH* is not necessarily related to the *pH* of the model calculation of the aqueous alteration.

An abundance of signals with repeated mass differences in the mass spectrum of SOM was reported by Naraoka et al. (2023, Science), providing evidence for a systematic reaction network including methylation, hydration, hydroxylation, and sulfation. Although we share the same hydrothermal fraction, the strongest signal in this fraction is polythionate, reflecting

the redox process of organic matter. The presence of elemental sulfur S₈, which Aponte et al. (2023, Earth Planets and Space) successfully detected, may be evidence of molecular evolution with progressive oxidation reactions in organic molecules after aqueous alteration. For example, monocarboxylic acid has been successfully detected in the Ryugu SOM (Naraoka et al., 2023, Science); monocarboxylic acid is weakly acidic, and we consider that the pH of the dissolved fractions also reflects these SOM characteristics.

Regarding the specific age of the reaction, the molecular functional groups present in the insoluble organic matter of the Ryugu sample indicate that the organic matter was affected by aqueous alteration in the Ryugu parent body, and no subsequent history of high-temperature impact heating has been reported (Yabuta et al., 2023, Science). That study concluded that the insoluble organic matter, which was mainly affected by aqueous alteration reactions, has not suffered significant alteration at Ryugu's surface (Yabuta et al., 2023, Science). Episodes of aqueous alteration and carbonate precipitation occurred between 2.5 and 5 Myr after CAI formation (Yokoyama+2023, Science). Soluble organic matter likely experienced a similar history, although subsequent molecular evolution is generally more likely to affect soluble organic matter than insoluble organic matter.

The water/rock ratio for aqueous denudation should be considered as the initial conditions, as indicated by Nakamura et al. (2023) (0.06–0.1 for least-altered and up to 0.9 for extensively altered lithologies); however, the question of whether the reaction conditions for organosulfur species can be considered as an extension of the fully reductive secondary mineral precipitation model is difficult to answer. The molecular evolution of the SOM may have been influenced by radical reactions, methylation and hydration, hydration, hydroxylation, and sulfurization, as well as general cosmic ray irradiation and solar heating (Naraoka et al., 2023, Science). We recognize that constraining the reaction conditions is a task to be accomplished as part of future work.

Other comments:

1. Online methods/ Ion chromatography: what is exchangeable carbonate?

→ This term was incorrect. The correct term is "exchangeable ions, carbonates,"

2. Fig 1b caption: Please define Exchangeable ions. What minerals are they associated with?

“We measured evaporitic salts (via #7-1 hot water extraction, see IDs in Extended Data Fig. 2 and Naraoka+2023a)”

Should it be Extended Data Fig. 1b?

→ We assume that the exchangeable ions are mainly associated with phyllosilicates. The figure citation has been corrected to Extended Data Fig. 1B, as suggested.

3. 1. 177-178 "The pH of the salt fraction at 24.6 °C (#7-1 in Supplementary Fig. 1) was weakly acidic..." why is there ref to this figure as no pH data is shown there?

→ This was a typographical error, and should have referred to Extended Data Figure 1, not Supplementary Figure 1. Thank you for pointing this out. We have changed the sentence.

4. Fig. 2 Please references where Ca, Mg, Na, and K were taken from for Orgueit, Tarda, Aguas Zarcas, and Jbilet Winselwan. If these samples were analyzed in this work, please add the data to Supplementary Table 1 "cosmic abundance (stars) also plotted for reference" Please add references

→ Orgueil, Tarda, Aguas Zarcas, and Jbilet Winselwan were measured in our study using the same protocol as Ryugu. For Tarda, Aguas Zarcas, and Jbilet Winselwan, the hot water (#7-1) and MeOH+DCM (#8) fractions were not allocated to the ion chromatography and ICP-MS analyses because they were preferentially used for other SOM analyses. Therefore, data for only the HCOOH (#9) and HCl (#10) fractions were available. Since these are the only data available, we have added descriptions to the Methods section and provided a data table (Supplementary Table 6). We also have added the reference.

5. Fig. 3 Dissolved carbonate is shown in (a) but it is unclear how it was detected. Was it measured with IC or was it a part of a different study? If the second, please add a reference

6. Suppl Table 2 " Note that dissolved inorganic carbon and silica were not measured"
HCO₃⁻ is reported in the Fig. 3a, please clarify

→ HCO₃⁻ cannot be measured by ion chromatography (IC) using a conductivity detector, but can be measured by using a hyphenated analysis of IC and Orbitrap-MS. However, the measurement of dissolved carbonate species in this study did not quantify the concentration of total dissolved inorganic carbon, so, as pointed out, dissolved carbonate could be misleading, and we have removed the information from the figure. Determination of dissolved carbonate species requires measurement of at least two of four parameters: alkalinity (TA), dissolved total carbonic acid concentration (TIC or DIC), pH, and pCO₂. In this case, only pH could be measured because the sample weight and solution volume were

too small for TA, TIC, and pCO₂ analyses.

Reviewer #3 (Remarks to the Author):

I would like to thank the authors for their detailed responses to my comments. All comments were fully addressed, and I recommend the paper to be accepted for publication.